

# The reproductive inhibitory effects of levonorgestrel, quinestrol, and EP-1 in Brandt's vole (*Lasiopodomys brandtii*)

Luye Shi[1,2,*], Xiujuan Li[1,*], Zhihong Ji[1], Zishi Wang[1], Yuhua Shi[1], Xiangyu Tian[1] and Zhenlong Wang[1]

[1] School of Life Sciences, Zhengzhou University, Zhengzhou, Henan, China
[2] School of Physical Education (Main Campus), Zhengzhou University, Zhengzhou, Henan, China
* These authors contributed equally to this work.

## ABSTRACT

**Background:** Rodent pests can inflict devastating impacts on agriculture and the environment, leading to significant economic damage associated with their high species diversity, reproductive rates and adaptability. Fertility control methods could indirectly control rodent pest populations as well as limit ecological consequences and environmental concerns caused by lethal chemical poisons. Brandt's voles, which are common rodent pests found in the grasslands of middle-eastern Inner Mongolia, eastern regions of Mongolia, and some regions of southern Russia, were assessed in the present study.

**Methods:** We evaluated the effects of a 2-mg/kg dose of levonorgestrel and quinestrol and a 1:1 mixture of the two (EP-1) on reproductive behavior as well as changes in the reproductive system, reproductive hormone levels, and toxicity in Brandt's voles.

**Results:** Our results revealed that all three fertility control agents can cause reproductive inhibition at a dosage of 2 mg/kg. However, quinestrol caused a greater degree of toxicity, as determined by visible liver damage and reduced expression of the detoxifying molecule CYP1A2. Of the remaining two fertility control agents, EP-1 was superior to levonorgestrel in inhibiting the secretion of follicle-stimulating hormone and causing reproductive inhibition. We believe that these findings could help promote the use of these fertility control agents and, in turn, reduce the use of chemical poisons and limit their detrimental ecological and environmental impacts.

## INTRODUCTION

Rodents are highly diverse, adaptable and have high reproductive rates (REF); such characteristics mean that several species of rodents are considered major pests because they cause devastating economic impacts to agriculture and the environment and are involved in spreading numerous diseases to humans and livestock (*Bordes, Blasdell & Morand, 2015*; *Capizzi, Bertolino & Mortelliti, 2014*; *Heroldová et al., 2012*; *Meerburg, Singleton & Kijlstra, 2009*; *Singleton, 2003*; *Singleton et al., 1999*; *Stenseth et al., 2003*). Besides, rodents hinder the attempts to alleviate poverty in Africa

Corresponding authors
Xiangyu Tian,
201531200038@mail.bnu.edu.cn
Zhenlong Wang, wzl@zzu.edu.cn

(*Mwanjabe, Sirima & Lusingu, 2002*) and Asia (*Singleton, 2003*). In Asia, a loss of 5% of total rice production, amounting to 30 million tons (sufficient to feed 180 million individuals for 12 months), has been reported (*Singleton, 2003*). Previous rodent management studies have primarily focused on mechanical controls (*Mayamba et al., 2019*), chemical rodenticides (*Goulois et al., 2017*), and biological controls (*Mayamba et al., 2019*; *Stenseth et al., 2003*). However, these prevention control measures can lead to ecological threats of their own (*Ettlin & Prentice, 2002*). In recent years, the Integrated Pest Management (IPM) and Ecologically Based Rodent Management (EBRM) strategies, which are based on ecological understanding, agronomy, environmental awareness and sociocultural considerations, have increasingly been accepted by individuals (*Altieri, Martin & Lewis, 1983*; *Krijger et al., 2017*; *Singleton et al., 1999*). Among the existing IPM and EBRM rodent control strategies, fertility control has been identified as a more appropriate, long-term control strategy compared with death by chemical poisoning because it is environmentally benign and humane (*Chambers, Singleton & Hinds, 1999*; *Liu et al., 2013*; *Shi et al., 2002*; *Tang et al., 2012a*, *2012b*; *Zhang, 2000*). This approach is consistent with the concept of non-toxic residues in EBRM strategy and can regulate population density by influencing the birth rate of pups (*Dell'Omo & Palmery, 2002*; *Fu et al., 2013*; *Liu et al., 2013*; *Liu, 2019*; *Su et al., 2016*).

Steroid hormone can cause infertility among animals by interfering with the normal functioning of the hypothalamic–pituitary–gonadal axis (*Chen, Su & Liu, 2017*; *Massawe et al., 2018*; *Su et al., 2016*, *2019*, *2015*; *Zhang, 2004*). Both levonorgestrel (L), a synthetic form of progesterone and quinestrol (Q), a synthetic estrogen, cause varying degrees of reproductive inhibition in a variety of rodents when combined (EP-1) (*Massawe et al., 2018*; *Zhang, 2004*). In recent years, research regarding the combination of the two agents in rodent pests has revealed the molecular mechanism associated with infertility, its environmental safety, and appropriate dosage (*Liu et al., 2012*; *Wang et al., 2011a*). Studies have shown that EP-1 reduces follicle-stimulating hormone levels and promotes luteinizing hormone levels, thereby inhibiting the maturation and ovulation of follicles in the ovary, ultimately leading to reduced fertility (*Lv & Shi, 2011*, *2012*). As a fertility control agent, EP-1 has the advantages of high reproductive inhibition, good palatability and strong sustainability. Moreover, it rapidly degrades and does not cause excessive environmental destruction (*Tang et al., 2012a*, *2012b*).

Although fertility control agents, such as oestrogen and triptolide, can cause temporary infertility in animals, they can be toxic (*Kejuan et al., 2007*; *Lipschutz et al., 1966*; *Maier & Herman, 2001*; *Van Aerts et al., 2019*; *Xu et al., 2019*; *Yuan et al., 2019*) and can cause death at higher dosages (*Lehmann et al., 1989*; *Zhang, Zeng & Lu, 2015*; *Zhang, 2015*). Therefore, it is important to estimate the toxicity in animals prior to using fertility control agents (*Ettlin & Prentice, 2002*; *Gao & Short, 1993*; *Ratti et al., 2015*; *Turner et al., 2011*; *Zhang, Zeng & Lu, 2015*). CYP1A2 is a member of the cytochrome P450 family that is involved in the metabolism of numerous drugs. In addition, it is responsible for the activation of precursors and carcinogens in the body (*Anttila, Raunio & Hakkola, 2011*; *Jiang et al., 2010*; *Zanger & Schwab, 2013*). Several factors, such as aging, genetics, diet, disease and toxic substances, can affect the metabolic activity of P450 enzymes

(*Stavropoulou, Pircalabioru & Bezirtzoglou, 2018*; *Zanger & Schwab, 2013*). Ingestion of drugs, can induce various transformations due to the action of microsomal cytochrome P450 and glucuronide transferase (*Jiang et al., 2010*; *Stavropoulou, Pircalabioru & Bezirtzoglou, 2018*). Therefore, as a detoxifying molecule, the activity of CYP1A2 may indicate toxicity levels in the body.

Brandt's voles (*Lasiopodomys brandtii*) are native to the grasslands of middle-eastern Inner Mongolia, eastern regions of Mongolia, and some regions of southern Russia (*Li et al., 2017*). They are considered pests because they can cause serious damage to grassland vegetation and grazing crops and can spread disease (*Zhang et al., 2003*). In the present study, Brandt's voles were treated with levonorgestrel, quinestrol, and a 1:1 mixture (EP-1) at a dosage of 2 mg/kg to determine the effects of each agent on reproductive status as well as their toxicity. During the experiment, the body weight of voles in different treatment groups was measured once every 7 days, and the animals were killed on Day 30 from the commencement of the treatment for measuring the effect of the steroids on the reproductive system and reproductive hormones and CYP1A2 levels in different treatment groups.

## MATERIALS AND METHODS

### Animal materials and experimental groups

Adult Brandt's voles were obtained from the Chinese Academy of Agricultural Science and were maintained in individual polycarbonate cages (37 × 26 × 17 cm) on a 14-h/10-h light/dark cycle at 20–24 °C and 35–50% humidity for at least 1 month. During the feeding period, voles were regularly fed with rat and rabbit feed (produced by the Henan experimental animal center, Zhengzhou, China) and fresh carrots.

To test the effect of various fertility control agents, 56 six-month-old healthy adult voles weighing between 40 g and 55 g were randomly divided into four groups (*n* = 14 in each group; 7 males, 7 females) as follows: L group (levonorgestrel), Q group (quinestrol), EP-1 group (1:1 mixture of levonorgestrel and quinestrol), and a control group (no fertility control agent; Table S1).

The experimental protocol was approved by the Animal Care and Use Committee of Zhengzhou University and was conducted in accordance with the Guide for the Care and Use of Laboratory Animals of China.

### Fertility control agent preparation and animal treatment

Levonorgestrel, quinestrol, and their 1:1 mixture (Dalian Meilun Biotechnology Co., Ltd., Dalian, China) were dissolved in 0.2 g/mL 2-hydroxypropyl-β-cyclodextrin (HPCD, Shandong Binzhou Zhiyuan Biotechnology Co., Ltd., Shandong, China). The final concentrations of the three fertility control agents were 0.5 mg/mL, and they were stored at 4 °C until required.

The treatment with the fertility control agents was performed for 7 days. To avoid the effects of different circadian rhythms on the animals, all experiments were conducted in the morning. During the experiment, animals in different treatment groups were orally gavaged with different fertility control agents at a dosage of 2 mg/kg using an elbow

gavage needle (No. 12 elbow gavage needle, Beijing Zhe Cheng Technology Co., Ltd., Beijing, China) at 9:00 A.M. daily. The dosage was based on previous studies by *Wang et al. (2011b)* and *Su et al. (2019)* where the dosage of levonorgestrel and quinestrol was 2 mg/kg and the dosage of EP-1 was 1 mg/kg of levonorgestrel and 1 mg/kg of quinestrol. Each experimental animal was separately housed during treatments, following which animals in each treatment group were housed in male–female pairs.

## Measurements of body weight, pregnancy rate, and number of fetuses

The weight of the experimental animals was measured five times: before injection with the fertility control agents and on days 7, 14, 21 and 28 from the commencement of the treatment.

All animals were killed 30 days after the start of treatment with an overdose of pentobarbital sodium (1%, one mL); female individuals in each group were dissected, and the pregnancy rate, number of fetuses and rate of fetal malformation were determined.

## Paraffin sections of the reproductive system and liver tissue

After the animals were killed, the testes and epididymides were removed from the male voles and the uteri and ovaries were removed from the female voles. Further, liver tissues of each animal were removed. Each organ was sequentially weighed and fixed in buffered paraformaldehyde (4%) for 24 h.

Paraffin sections of each reproductive organ and liver tissues were sequentially prepared and stained with hematoxylin and eosin, and the prepared sections were assessed using a microscope with a magnification of ×100. Five fields of view were randomly selected for histological observation and photographing.

## Determination of estradiol, follicle-stimulating hormone, testosterone, and corticosterone levels

Sub-orbital sinus blood was collected on Day 30. The blood was stored at room temperature for 20–30 min to clot, and serum was extracted from the blood via centrifugation (3,000 rpm, 20 min) and was transferred into a −20 °C freezer. The serum levels of estradiol, follicle-stimulating hormone, testosterone, and corticosterone were measured using the E2 ELISA kit, FSH ELISA kit, T ELISA kit and CORT ELISA kit (Shanghai MLBIO Biotechnology Co., Ltd., Shanghai, China), respectively. All procedures were performed in accordance with the manufacturer's instructions.

## Total RNA extraction and quantitative real-time PCR

Total RNA was extracted from the brain tissue of each animal using TRIzol reagent (Invitrogen, Carlsbad, CA, USA), in accordance with the manufacturer's instructions. Residual DNA was removed by treatment with RNase-free DNase I (Takara Bio, Dalian, China). RNA integrity was verified using agarose gel electrophoresis (1.2%), and RNA concentrations were measured using an Agilent 2100 Bioanalyzer (Agilent Technologies, Santa Clara, CA, USA).

**Table 1 Details of the real-timePCR primers used to detect expression levels of CYP1A2.**

| Gene name | Sequences | Length (bp) | Annealing temperature | PCR efficiency |
|---|---|---|---|---|
| *CYP1A2* | Forward: TCGTCCTCTTGCTACTTA | 1,177 bp | 55 °C | 0.91 ± 0.02 |
| | Reverse: TCGTCCTCTTGCTACTTA | | | |

The total RNA extracted from the brain tissues was reverse-transcribed into cDNA using a reverse transcription kit (Takara Bio, Shiga, Japan). The obtained cDNA was stored at −20 °C prior to real-time PCR.

The CYP1A2 sequences of the near relatives of Brandt's voles were determined and compared on National Center for Biotechnology Information before designing primers for real-time PCR using Primer3 online software (*Untergasser et al., 2012*). The primers were further verified via sequencing after the PCR products were assembled into cloning vectors. Details of primer sequences are shown in Table 1.

Real-time PCR was performed on a Rotor-Gene 3000 fluorescence quantitative PCR instrument (Corbett Research, Leipzig, Germany) using a default reaction procedure configuration. The reaction volume was 20 μL and included 0.5 μL each of specific forward and reverse primers (20 μM), two μL of the diluted cDNA template, one μL of the PrimeScript RT Enzyme Mix, six μL of the 5× PrimeScript Buffer and 10 μL of RNase-free $H_2O$. β-actin was used as the internal control gene (forward: GTCGTACCACTGGC ATTGTG; reverse: CCATCTCTTGCTCGAAGTCC), and the relative gene expression was determined using the comparative CT method (*Pfaffl, 2001*).

### Statistics analysis

All data were analyzed using SPSS (version 22.0, SPSS Inc., Chicago, IL, USA) and presented as mean ± SE. A *p* value of <0.05 denoted statistical significance. A generalized linear model with repeated measurements was used to analyze the rate of change of body weight, and two-way ANOVA was used to analyze the effects of the fertility control agents on the reproductive system, hormone levels and gene expression level in each group.

## RESULTS

### Effects of fertility control agents on the body weight

During the experiment, the body weight of the L treatment group increased during the first 21 days following which it began to slowly decrease. The remaining three treatment groups (control, Q and EP-1) initially showed decreases in body weight but subsequently increased (Fig. 1; Table S1). The maximum weight loss in the Q group occurred by Day 14 of the experiment and the weight was significantly lower in this group than that in the other three groups ($p = 0.031$). This indicated that quinestrol significantly reduced the body weight of Brandt's voles ($p = 0.011$) but levonorgestrel and EP-1 did not ($p = 0.082$, $p = 0.113$, respectively).

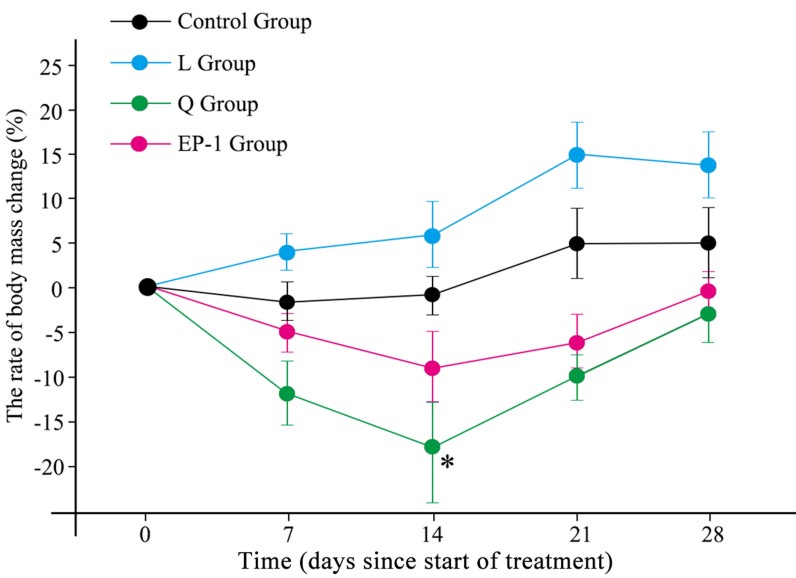

**Figure 1 Body mass changes of voles in the three fertility control agent-treated groups and control group.** The asterisk (*) represents a significant difference from the control group.

**Table 2 The number of fetuses among the four groups of voles at Day 30 of the experiment.**

| Group | Pairing number | Pregnancy rate (%) | Number fetuses per pregnant female |
|---|---|---|---|
| Control | 7 | 85.71% (6) | 8, 7, 2, 9, 7, 1 |
| Quinestrol | 7 | 0 | 0 |
| Levonorgestrel | 7 | 14.29 (1) | 2 |
| EP-1 | 7 | 0 | 0 |

## Effects of fertility control agents on reproduction

On Day 30 of the experiment, 6 of the 7 females in the control group were pregnant, with a pregnancy rate of 85.7%. However, among the three treated groups, only one female in the L group experienced a failed pregnancy, and none of the other females showed any signs of pregnancy (Table 2). All embryos in the control group showed normal development, whereas the two embryos in the L group were both resorbed fetuses.

## Effects of fertility control agents on the reproductive system

Compared with the control group, L ($p = 0.03$), Q ($p = 0.02$), and EP-1 ($p = 0.02$) demonstrated a significantly reduced uterine weight in the female voles (Fig. 2A). Furthermore, the results of the morphological observations showed that the uterine walls in females in the three treatment groups were thinner than those in females in the control group (Fig. S1). However, the three fertility control agents had no significant effect on the ovarian weight of the female voles ($p = 0.81$ for L, $p = 0.93$ for Q and $p = 0.79$

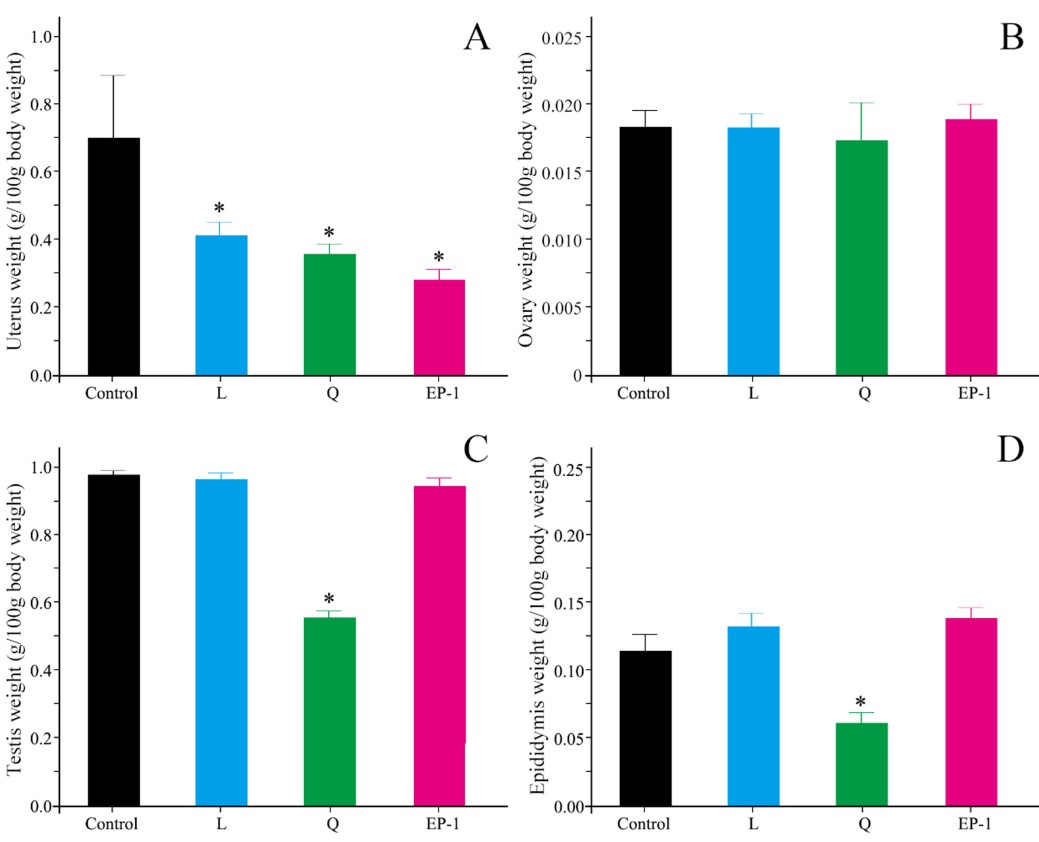

**Figure 2 Uterine weight (A) ovarian weight (B) testicular weight (C) and epididymial weight (D) in the four groups of voles at Day 30 of the experiment.** Values are presented as mean ± SE. The asterisk (*) represents a significant difference from the control group.

for EP-1; Fig. 2B). In addition, no significant difference was observed in the development of ovarian tissue in the treatment groups (Fig. S2).

Compared with the males in the control group, the testicular shape and volume in males in the L and EP-1 groups did not show significant changes, whereas the testes of a few males in the Q group showed a darker color and decreased volume (Fig. S3). In addition, compared with the other three groups, males in the Q group showed a significant decrease in their testicular weight by approximately 40% ($p = 0.03$, Fig. 2C), a sparser seminiferous tubule, acute apoptosis in sperm cells and testicular stromal and supporting cells at all levels (Fig. S4). Similar to the findings from the morphological observations, compared with the control group, the epididymal weight of the individuals in the Q group decreased by 50.6%, with extremely few individuals experiencing unilateral epididymal atrophy ($p = 0.01$, Fig. 2D).

## Effects of fertility control agents on the reproductive hormones

No significant differences were observed in estradiol, testosterone, or corticosterone levels in the animals among different treatment groups (Fig. 3). However, the

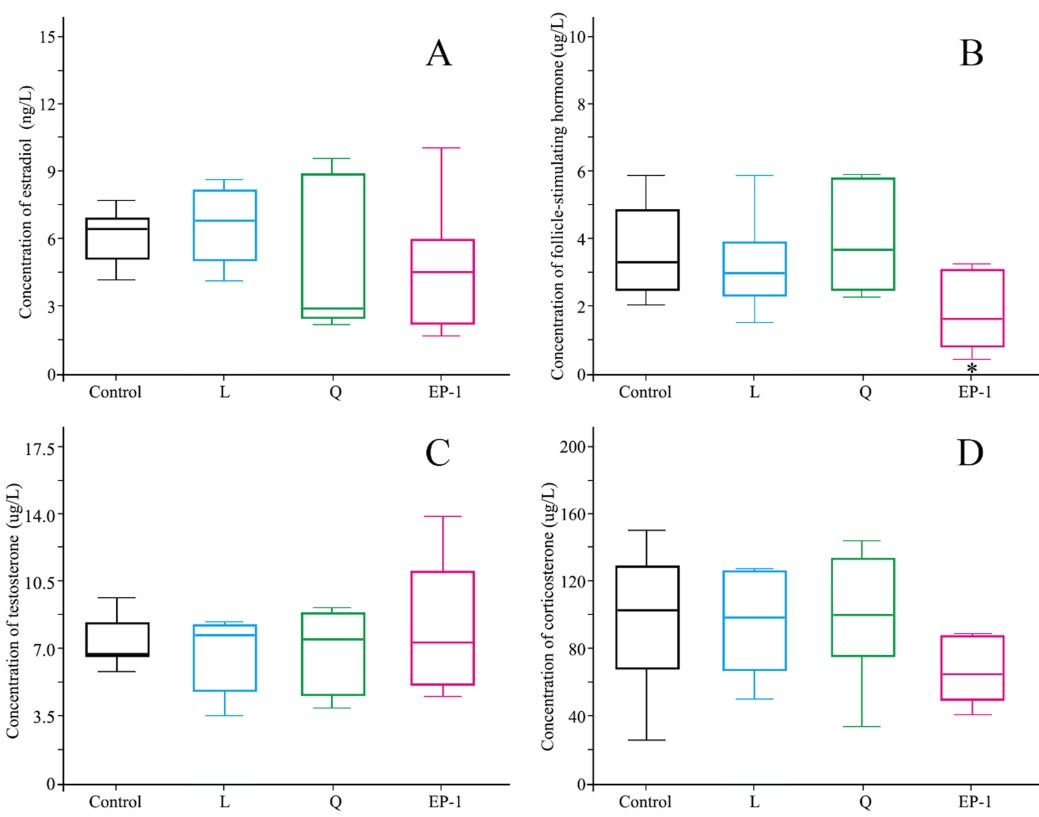

**Figure 3 Levels of estradiol (A) follicle-stimulating hormone (B) testosterone (C) and corticosterone (D) in the four groups of voles on Day 30 of the experiment.** Estradiol and follicle-stimulating hormone were measured only in females, and testosterone and corticosterone were measured only in males. Each group of data is represented by a box plot. In the group of data, the top and bottom edges outside the box plot represent the maximum and minimum values, respectively, and the horizontal lines in the box plot represent the median of the set of data. The asterisk (*) represents a significant difference from the control group.

follicle-stimulating hormone levels in the L ($p = 0.03$) and EP-1 ($p = 0.01$) groups were significantly lower than those in the control group (Fig. 3).

## Effects of fertility control agents on the expression of CYP1A2

Compared with the control group, the expression of CYP1A2 in the other three groups was significantly reduced. Specifically, the expression levels of CYP1A2 in the L, Q and EP-1 groups were reduced by 40%, 61% and 43%, respectively (Fig. 4A).

## Effects of fertility control agents on liver tissue

Following treatment, the liver tissue in animals treated with quinestrol appeared dark in color and lost its normal toughness. Furthermore, a small number of EP-1-treated animals showed the above mentioned liver morphological changes, whereas no liver morphological changes were observed in the L or control groups.

From the liver tissue sections, the hepatocytes of the control, L and EP-1 groups showed normal development, whereas those of the Q group showed tissue edema and local bleeding (Fig. 4B).

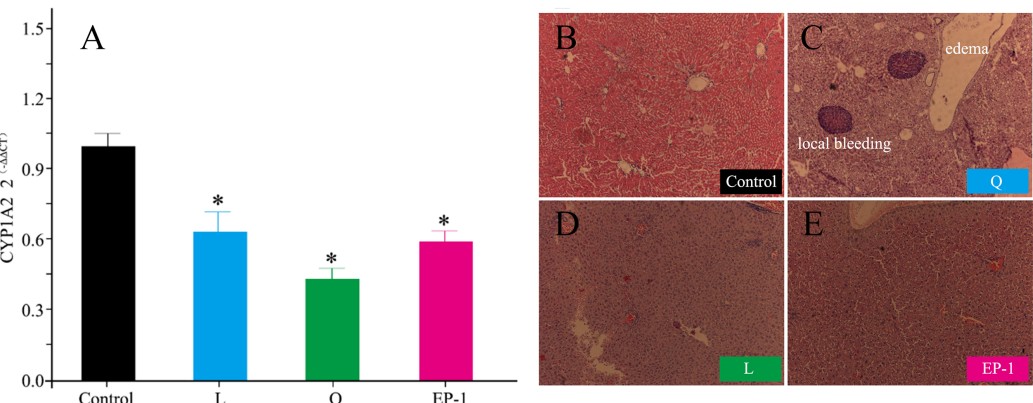

**Figure 4 Gene expression levels of CYP1A2 (A) and liver histological features (×100) at Day 30 after commencement of the control group (B) and treatment with Q (C) and L (D) and EP-1 (E).** The liver tissues in (B) are visualized under the original magnifications of hematoxylin and eosin staining.

# DISCUSSION

Currently, chemical poisoning using agents such as anticoagulant rodenticides and zinc phosphide remains one of the most important methods for rodent pest control (*Haim et al., 2007*; *Labuschagne et al., 2016*; *Liu, 2019*). EBRM has recently been implemented in several countries, such as Vietnam and Indonesia as well as in the African continent and mainly involves the use of trap–barrier systems, community rat hunting campaigns and habitat management (*Brown et al., 2005*; *Brown & Khamphoukeo, 2007*; *Jacob, Singleton & Hinds, 2008*; *Massawe et al., 2011*; *Singleton, Jacob & Krebs, 2005*; *Wang et al., 2017*). The addition of fertility control as an ecologically friendly method has not yet been widely possible. In the present study, we evaluated the reproductive inhibitory effect of quinestrol, levonorgestrel and EP-1 in Brandt's voles. Our findings suggest that the effects of these fertility control agents require further research so that they can be applied as part of an integrated approach in future large-scale applications.

The study results showed that quinestrol treatment led to significantly reduced body weight of voles; however, levonorgestrel and EP-1 treatment had no significant effects, suggesting that quinestrol is more toxic than the remaining two fertility control agents. Appropriate fertility control agents should cause competitive reproduction interference without affecting the animal's normal development and other biological functions. Therefore, any treatment administered should avoid killing the pests, and the doses should not negatively affect their growth and development (*Liu et al., 2018*; *Turner et al., 2011*). In previous studies, *Su et al. (2019)* found that EP-1 in different ratios (1:1, 2:1, 1:2) can reduce the sperm density and vitality of male mice and cause atrophy of the epididymis and seminal vesicles; on the other hand, the 1:1 ratio of EP-1 had the best effect on reproductive inhibition. In addition, *Wang et al. (2015)* proved that 2 mg/kg of quinestrol had the best reproductive inhibition effect on Brandt's voles by comparing different concentrations of infertility reproductive inhibitory effects and could sustain for more than 90 days. In our study, we referred to these experiences to select the reproduction inhibitor

dose and EP-1 ratio. The results showed that a dose concentration of 2 mg/kg and EP-1 ratio of 1:1 were both appropriate, significantly inhibiting the reproductive process of Brandt's voles without affecting the normal developmental of animals. However, levonorgestrel and EP-1 appeared less toxic to the animals than quinestrol.

The reproductive behavior analysis indicated that all three fertility control agents successfully sterilized the Brandt's voles. The uterine weight (g/100 g body weight) of females treated with all three fertility control agents was significantly lower than that of the females in the control group, suggesting that the three fertility control agents had inhibited reproduction. In addition to the effect on the uterine weight, the testicular and epididymal weight (g/100 g body weight) of the males in the Q group were significantly lower than those in the control group, suggesting that quinestrol inhibits reproduction in both female and male Brandt's vole individuals. However, although the uterine walls of the females in the control group were thinner than those in the control group, this may be due to the fact that the females in the control group experienced pregnancy, so the thickness of the uterine wall could not be used to judge the effect of the sterility agents. Previous research has shown that quinestrol is a stable estradiol homologue that is stored in the adipose tissue and can be slowly released (*Zhao et al., 2007*). Quinestrol primarily inhibits the release of gonadotropin releasing hormone from the hypothalamus, thereby inhibiting follicle growth, whereas levonorgestrel primarily inhibits the release of follicle-stimulating hormone from the pituitary gland to block ovulation and disrupts the proliferation and differentiation of endometrial cells, preventing embryonic implantation (*Zhao et al., 2007*).

Previous studies in mice and humans have shown that fertility control agents can inhibit pregnancy by interfering with the synthesis of follicle-stimulating hormone, estradiol, testosterone, corticosterone, and other reproductive hormones (*Spona & Huber, 1987*). In the present study, 30 days after the commencement of fertility control agent treatments, no significant difference was observed in the estradiol, testosterone, or corticosterone hormone levels. However, we observed a significant decrease in follicle-stimulating hormone levels in the EP-1-treated group; this phenomenon can be attributed to mixed levonorgestrel and quinestrol interfering with the in vivo regulation of the hypothalamic–pituitary–adrenal axis (*Bednarek, 2007*; *Likis, 2002*). However, the specific interference mechanism of EP-1 and why levonorgestrel and quinestrol do not inhibit follicle-stimulating hormone levels when used alone still require further research.

CYP1A2 is a member of the cytochrome P450 enzyme system, which mediates over 90% of the drug transformation process in vivo (*Porter & Coon, 1991*). Recently, research on CYP1A2-mediated drug metabolism has shown that several drugs can cause toxic side effects on the body by inhibiting the activity of CYP1A2 (*Faber, Jetter & Fuhr, 2005*; *Huang et al., 2014*; *Johnson et al., 2012*; *Li et al., 2018*; *Martinez et al., 2013*; *Wei et al., 2018*). We observed that all three fertility control agents inhibited the expression of CYP1A2 in Brandt's voles. Among the three fertility control agents, the strongest inhibitory effect was observed in the Q group, which was further confirmed by the observation of liver sections, indicating that quinestrol is more toxic than the other two fertility control agents.

In summary, our results show that all three fertility control agents can cause reproductive inhibition in Brandt's voles at a dosage of 2 mg/kg. However, due to the limitation of the number of animals and experimental conditions, we did not pair the animals of the reproductive inhibition treatment groups with healthy and untreated individuals in the matching process after gavage, making it impossible to distinguish whether reproductive inhibition was caused by the female, male, or both in subsequent analysis. Our current findings suggest that EP-1 was superior to levonorgestrel in inhibiting the secretion of follicle-stimulating hormone and causing reproductive inhibition. Furthermore, compared with the other two agents, quinestrol caused a greater degree of toxicity. Therefore, EP-1 administered at a dosage of 2 mg/kg appears adequate to inhibit the reproduction of Brandt's voles.

## CONCLUSION

In previous studies, both levonorgestrel and quinestrol were demonstrated to have inhibitory effects on reproduction in various rodents (*Huo, Shi & Wang, 2007*; *Liu et al., 2017*; *Lv & Shi, 2011*, *2012*; *Wan et al., 2006*). However, the use of biological fertility control agents for pest control has not yet been widely implemented. One possible reason for this is that specific dosages and side effects have not yet been established. In the present study, we have elaborated the effects of the three fertility control agents on the reproductive system, reproductive hormone levels, CYP1A2 expression level, and liver damage. We determined that EP-1 was the most suitable fertility control agents for the reproductive inhibition of Brandt's voles. We hope that these results will help to promote the field use of these fertility control agents so that they can effectively reduce rodent pest populations as well as limit the ecological damage to the environment associated with chemical poisons.

## ACKNOWLEDGEMENTS

Besides the funding support, we thank Shiming Gu, Mengke Li, Xuqin Wang, Yue Wu, and Mengyang Li for their help in feeding the experimental animals.

### Funding

This work was supported by the National Natural Science Foundation of China (grant no. 31372193). The funders had no role in study design, data collection and analysis, decision to publish, or preparation of the manuscript.

### Grant Disclosures

The following grant information was disclosed by the authors:
National Natural Science Foundation of China: 31372193.

### Competing Interests

The authors declare that they have no competing interests.

## Author Contributions

- Luye Shi performed the experiments, analyzed the data, prepared figures and/or tables, authored or reviewed drafts of the paper, and approved the final draft.
- Xiujuan Li performed the experiments, authored or reviewed drafts of the paper, and approved the final draft.
- Zhihong Ji performed the experiments, analyzed the data, prepared figures and/or tables, and approved the final draft.
- Zishi Wang analyzed the data, authored or reviewed drafts of the paper, and approved the final draft.
- Yuhua Shi performed the experiments, prepared figures and/or tables, and approved the final draft.
- Xiangyu Tian analyzed the data, prepared figures and/or tables, and approved the final draft.
- Zhenlong Wang conceived and designed the experiments, authored or reviewed drafts of the paper, and approved the final draft.

## Animal Ethics

The following information was supplied relating to ethical approvals (i.e., approving body and any reference numbers):

The experimental protocol was approved by the Animal Care and Use Committee of Zhengzhou University and is in accordance with the Guide for the Care and Use of Laboratory Animals of China.

## Data Availability

The CYP1A2 sequence dataset is available at GenBank: MT501643.

## Supplemental Information

Supplemental information for this article can be found online at http://dx.doi.org/10.7717/peerj.9140#supplemental-information.

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
