# Peer review of "The reproductive inhibitory effects of levonorgestrel, quinestrol, and EP-1 in Brandt’s vole (Lasiopodomys brandtii)"

_PeerJ, doi:10.7717/peerj.9140_

## Round 0.1 · original submission · Major Revisions

Two reviewers have now read your manuscript. Both find this study to be well done and have merit. That being said, there are a substantial number of revisions to address.

In particular, there are a number of revisions with respect to the reporting of methods and results that both reviewers have identified. I encourage the authors to pay close attention to the reviewers' comments about reporting when revising your manuscript. Please note I have several comments of my own below as well.

Line 51: how does fertility control influence sex ratio?

In the methods for the paraffin sectioning, please state what features were being assessed in both the liver and reproductive organs.

In figure 3, the y-axes incorrectly say “ecoefficient”. Also, please add a description of what this coefficient is in the figure legend. In panel C, “ovarian” should be “ovary”

In figure 4, all y-axes labels are in different units (ng, nmol, U, g) per volume. Please standardize these more.

I look forward to receiving a revised version of this study.

Reviewer 1 ·

Basic reporting

This manuscript describes an assessment of the synthetic steroids, Levonorgestrel (L) and Quinestrol (E) and their combination known as EP-1 (1:1). The doses were delivered for 7 days and animals assess for the effects on reproductive parameters at Day 8, 15, 21 and 30 days from the start of treatment.
The paper requires major revisions and presentation of more detailed experimental methods and results.
English expression requires some improvement.
Some use of terminology is ill-defined or confusing - also see general comments below.
The author’s use of the term “sterility” should be defined and justified. The fertility control agents (E, P and EP-1) generally have a short term effect on fertility at the doses being used – thus animals become infertile for a period of time, but they do not become sterile. Thus these synthetic steroids should be called fertility control agents because they induce temporary and reversible infertility.
Reference to the relevant literature is not comprehensive and not always pertinent to their statements.
It would not be possible to repeat this study without considerably more information.

Experimental design

Research question is defined but some elements not well justified - for example why specific treatment dose of 2mg/kg was chosen given background literature.
Methods not well described - this study could not be easily replicated.
Also no housekeeping gene expression was assessed in parallel with CYP1A2 expression.
See general comments below

Validity of the findings

Nott all underlying data was provided - e.g. testis measurements and volumes.

Conclusions not well justified with respect to literature.

Additional comments

In parts, the English expression and logic requires some improvement.
For example the first sentence (lines 39-41) could be more logically expressed. “Rodents are highly diverse, adaptable and have high reproductive rates such that many species are considered major pests because they cause devastating economic impacts to agriculture and the environment (multiple references should be included here, including Singleton et al 2010; Singleton et al 1999).
Lines 42 – onwards. Reference should also be made to ecologically-based rodent management (ECRBM) – this is a more comprehensive approach than traditional IPM or the use of single strategies such as a rodenticides, trapping, fumigating etc. See Singleton et al 1999 for original references. And further references in Singleton et al 2010.

Line 47-50: The authors are incorrect to state “that fertility control is currently one of the most widely used IPM strategies…”. There have been no broad scale applications of fertility control for rodents – there have been many experimental assessments and many proposed uses of fertility control for rodent management, but it is not widely applicable at this stage for any fertility control agents. EP-1 is under test for some species under field conditions but little efficacy has not been reported in the literature. Authors should refer to Liu et al 2012 Effects of Quinestrol and Levonorgestrel on populations of plateau pika…... Pest Management Science, 68, 592-601.
Lin 50: “cubs” should be “pups” .
Line 55-56; the only relevant reference for this statement is Liu et al 2017. It is the only reference that deals with fertility control agents. Dempster et al (1991) (note the year of publication is incorrectly quoted in the reference list) were testing vocalisations and their importance for communication. Ma et al (2019) is an assessment of an alternative rodenticide.

Line 68: Add references to this statement. It is very general - however, the authors would appear to be referring to synthetic steroids only??

Line 85: Suggest change “sterility” to “reproductive status”
Line 87: Change to read “were quantified at 7 days intervals for 30 days from the beginning of treatment.

Line 97: Please indicate adult body weights and their range for males and females.
Line 109: Please clarify the preparation of the concentrations of the respective steroids.
When tested as E or P alone were they delivered at a concentration of 0.5mg/ml and 2mg/kg body weight?
When delivered as EP-1 did the solution comprise 0.5 mg/mL for E plus 0.5 mg/mL P? If so, please define what you mean by 2mg/kg mean? Is it 2mg/kg of each steroid?
Please add a justification for why this concentration was chosen. Refer to earlier literature for Brandt’s voles and other rodents.
Line 112: Animals were not “injected”, they were orally gavaged using a gavage needle (please add size of gavage needle used). Please indicate range of volumes that were delivered by oral gavage.
Line 115: Please clarify the number of male-female pairs - I think it is 7 per treatment group? If so then this pairing arrangement compromises the interpretation of the results as the outcome of mating (pregnancy rates) may have been due to infertility in the male, or female, or both.

Line 119: Generally, the convention is to express results with respect to the start of treatment which is designated as Day 0, not as days from the end of treatment. Therefore please change these days to Day 8, 15, 22 and 30 from start of treatment? Some of the results are presented as time since start of treatment (see Figure 1, Day 0, 7, 14, 21, 28).
Line 124: Change “sacrificed” to “killed”. Please add how the animals were humanely killed.
Line 126: What concentration of paraformaldehyde (4%??) and was it buffered?
Line 134: Were samples taken under light anaesthesia prior to killing of the animal for tissues etc?
Line 139-140: Were quality control samples added to each assay run? What were they and what was the variation observed within and between assays?
Line 142: Was the expression of any housekeeping genes in the brain also assessed? This is an important control for the method.
Line 160: What is Section 2.4? Is this a part of a thesis?
Line 184: The one female in the Levonorgestrel group that was pregnant, showed resorptions – therefore it was a failed pregnancy, not a prolonged gestation as no fetuses were ever delivered. What size were the resorption sites - small or large? What stage of development were the foetuses.
Line 185: The number of fetuses per animal should be presented as using the number that were pregnant (i.e. 6 in the control group and 1 in the L group). Using n=7 to calculate an average artificially reduces litter size for the controls.
Suggest the data is presented in a table.

Group n Pregnancy rate (%) (n) Number fetuses per pregnant female
Control 7 87.5 (6) 5, 6, 5, 4, 7, etc
Quinestrol 7 0 0
Levonorgestrel 7 14.3 (1) How many resorption sites were present?
EP-1 7 0 0

Line 186: The correct term here is “resorbed” not “absorbed”

The use of “coefficient” here is confusing and unnecessary. The text can be simplified by stating that the uterine weight (g/ 100g body weight) was significantly reduced…. Similarly for the ovarian weights.
Line 192: Please define what you mean by “anatomical” observations. Are these gross observations of fresh tissues or the histological observations of the uterus? Were actual measurements taken to show difference of is it an assertion?

Line 195-196: Was histology of the ovaries performed? Did all ovaries from treated groups show similar numbers and types of follicles – did they have any new ovulations/corpora lutea? And were they different from the control ovaries?
Line 197-199: Please define testicular “shape” and provide the actual data for the testis volumes. The latter requires that the length and width of each dissected testis was measured.
Line 200: Delete co-efficient - use testicular weight (g/100g body weight).
Line 202: Define what is meant by “anatomical” observations?
Line 203: Delete co-efficient - use epididymal weight (g/100g body weight).
Were seminal vesicles weighed and examined?
Line 213: Was the expression of a basic housekeeping gene also assessed for the brain tissues as well as CYP1A2?
Line 221: I do not understand what is meant by “eroded and filled with water. It also became blackened and brittle and lost its normal toughness.” Were these gross observations made at the time of dissection? The histology of the tissue (Figure 5B) and its description does not convey an impression of such major damage. Please clarify.
Lines 229: Please acknowledge that ECRBM has been applied in several counties (Vietnam, Indonesia, countries in the African continent) in recent years (see Singleton et al 2010 and papers within that book).

Line 237: delete “ontogeny” and state that normal development and other biological functions should not be impaired when using a fertility control agent. Please consider re-drafting this sentence as the meaning is not very clear to this reader. Fertility or sterility control agents are not meant to disrupt social behaviours.
Line 244 – 251: this paragraph is essentially a repeat description of the results. Also please note the earlier comment that the pairing of treated animals of each sex does not elucidate the mechanism of effect. Reference should be made to the fact that the administration of steroids disrupts the normal endocrine feedback loop between the hypothalamus- pituitary and gonads, not necessarily a major effect on the adrenal axis. The effects are not generally permanent – that is the effect of treatment is reversible over time.
Line 277: change “organisms” to “rodents”.

Figures – in general figure legends need more explanation and should be stand alone form the text.
Figure 1: Presentation of actual body weights for each group rather than loss of body weight would be more useful. Please indicate that the body mass changes are Mean ± SE. The x-axis should be described as Time (days since start of treatment). Animals were killed on Day 30 so is this axis correct?

Figure 2: Not needed if data is presented more comprehensively in table format.

Figure 3: Please change y-axes to read as Uterine weight (g/100 g body weight); Ovarian weight (mg/100 g body weight); Testis weight (g /100 g body weight) and Epididymis weight (g/100 g body weight. Add “Values are Mean ± SE.”
Also please confirm whether the values presented for the ovary, testis and epididymis represent are the combined weights (total) of these organs corrected for body weight.

Figure 4: Please confirm that the units used for each hormone are correct – especially for corticosterone (g/ml??). It is hard to believe that the FSH values for the Levonorgestrel group are significantly different from the controls. Please explain your box plots.
Figure 5: See earlier question regarding expression of housekeeping genes relative to CYP1A2.
The liver sections need to be enlarged - and the magnification added.

References
Please carefully check that all references are correct. Below are two examples where incorrect or insufficient information is provided.
Line 302: The Dempster et al article was published in 1991 not 2010.
Line 400: Which journal for Su et al (2016)?

The authors should also consider several other references which are pertinent to the topic.

Singleton GR, Brown PR, Jacob J, Aplin KP (2007) Unwanted and unintended effects of culling: a case for ecologically-based rodent management. Integr Zool 2:247–259
Singleton GR, Belmain SR, Brown PR, Hardy B (editors) (2010) Rodent outbreaks: ecology and impacts. International Rice Research Institute, Los Banos, 289 pp. Specific papers in this book could should be cited as well as the entire book?
Nghiem LTP, Soliman T, Yeo DCJ, Tan HTW, Evans TA, Mumford JD, Keller, RP, Baker RHA, Corlett RT and Carrasco LR (2013). Economic and environmental impacts of harmful non-indigenous species in Southeast Asia PLOS One, 8 (8), e 71255, 1-9.
Zhang Z (2015) A review on anti-fertility effects of levonorgestrel and quinestrol (EP -1) compounds and its components on small rodents. Acta Theriol Sin 35:203–210
Zhao M, Liu M, Li D et al (2007) Anti-fertility effect of levonorgestrel and quinestrol in Brandt’s voles (Lasiopodomys brandtii). Integr Zool 2:260–268

Qu, Lui, Yang, Zhang and Zhang (2015). Effect of fertility control in plateaus pikas……. Acta Theriologica Sinica 35 (2), 164-169.

Zhang, Wang, Liu, Qu, Liu, Zhang and Zhao (2014). Degradation of the potential rodent contraceptive quinestrol and elimination of its estrogenic activity in soil and water. Environ Sci Pollut Res. 21, 652-659.

Reviewer 2 ·

Basic reporting

no comment

Experimental design

The authors claimed that paraffin sections and the H&E staining were performed in the reproduction organs of Brandt’s voles (line 123-130), but the description for the histology variation of reproductive organs was not found in the Result section.

The gavage of 2 mg/kg EP-1 was composed of 1 mg/kg levonorgestrel and 1 mg/kg quinestrol (totally 2 mg/kg), or 2 mg/kg levonorgestrel and 2 mg/kg quinestrol (totally 4 mg/kg)?

Validity of the findings

Some hormone measurements were usually performed only in males or females. For example, testosterone measurement was usually performed only for males, while estradiol measurement was usually performed only for females. In Figure 4, the authors should claim the animal sex (male, female or both) of the measurements for the 4 hormones (estradiol, FSH, testosterone, and corticosterone).

The authors should supply high-resolution photographs for Figure 5B.

Additional comments

The authors studied the anti-fertility effect of 2 mg/kg levonorgestrel, quinestrol, and EP-1 in Brandt’s voles. They found that the treatment by the 3 anti-fertility reagent above all induced infertility successfully, and the expression of CYP1A2 in liver significantly decreased after the treatment. The study supplied valuable data for the anti-fertility effect of EP-1 in Brandt’s voles. However, some points were still needed to be improved (see the 3 areas above).

---

## Round 0.2 · Minor Revisions

Thank you for your thorough revisions and responses to the reviewers.

The reviewers have identified several issues that remain to be addressed. In particular, I emphasize that Reviewer 1 has highlighted some issues with the interpretation of your results and limitation of your study that you need to discuss in your manuscript.

In your revision, please transcribe the comments Reviewer 1 provided in the attached PDF and provide a response to each comment.

I look forward to receiving a revised version of your study.

Reviewer 1 ·

Basic reporting

English expression has been improved. The authors have endeavoured to address the various reviewer comments made on their first submission. However, there are still several issues that need to be clarified and addressed. I have noted many detailed comments on the manuscript.

Experimental design

The pairing of males and females confounds the responses observed in terms of rates of pregnancy. The lack of pregnancy may have been due to either or both sexes being infertile due to the treatment. The authors should address this limitation in their methods and Discussion.
The absence of the analysis of an appropriate house-keeping gene to validate differences in expression of CYP1A2 requires addressing.

Validity of the findings

Some concerns about author's interpretations of dosing effects.

Lack of use of a stable house-keeping gene to validate differences in expression of CYP1A2 requires clarification.

Additional comments

See attached document

Annotated reviews are not available for download in order to protect the identity of reviewers who chose to remain anonymous.

Reviewer 2 ·

Basic reporting

No comment.

Experimental design

No comment.

Validity of the findings

For Figure 5B, I am not sure whether the dark areas represented local bleeding. The authors should show a higher magnification inset in which people can observe whether the dark areas were composed of red blood cells.

Additional comments

In general, the manuscript has been improved a lot. However, there is still one point to be revised (see above).

---

## Round 0.3 · accepted · Accept

Thank you for your thorough revision and thoughtful responses to the reviewers. This manuscript is greatly improved and a clear presentation of your study. I am excited to accept your manuscript for publication.

Contingent upon acceptance is a final revision in accordance with one of the reviewer's comments - please also change the word "sterility" in your abstract, the supplemental figure legends, and on line 276.